# Seeing the Wind from a Falling Leaf

Zhiyuan Gao[1][*]    Jiageng Mao[1][*]    Hong-Xing Yu[2]    Haozhe Lou[1]    Emily Yue-Ting Jia[1]
Jernej Barbic[1]    Jiajun Wu[2]    Yue Wang[1]
[1]University of Southern California    [2]Stanford University
{gaozhiyu, jiagengm, haozhelo, eyjia, jnb, yue.w}@usc.edu
{koven, jiajunwu}@cs.stanford.edu

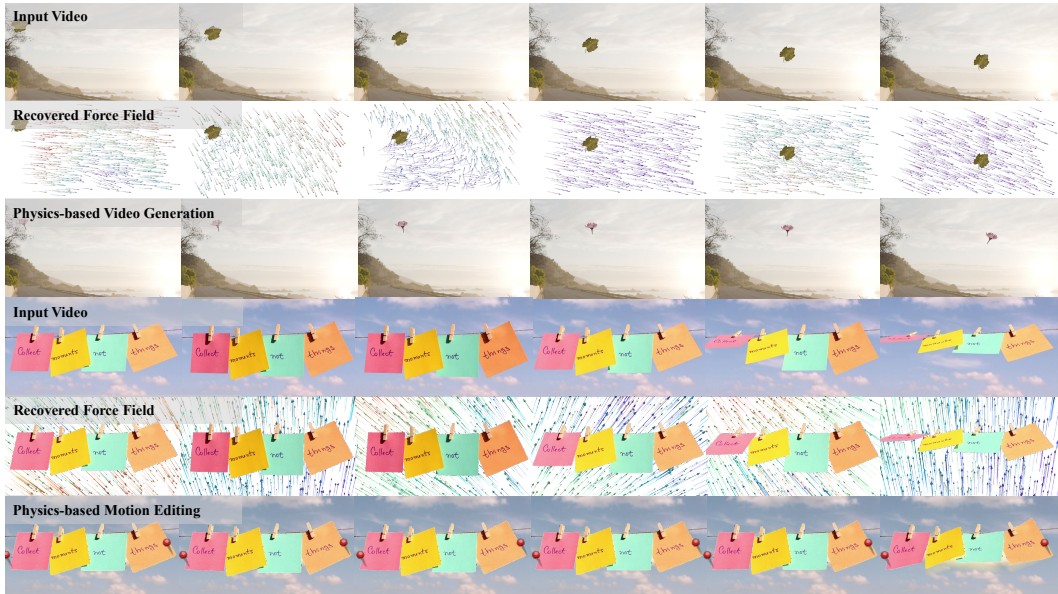

Figure 1: We propose an end-to-end differentiable framework capable of estimating invisible forces directly from video data, mimicking the human ability to perceive unseen physical effects through vision alone. This approach enables applications such as physics-based video generation, where new objects can be seamlessly introduced into a scene and simulated within the same force field. Force strength: from low to high (best viewed in colors).

## Abstract

A longstanding goal in computer vision is to model motions from videos, while the representations behind motions, i.e. the invisible physical interactions that cause objects to deform and move, remain largely unexplored. In this paper, we study how to recover the invisible forces from visual observations, e.g., estimating the wind field by observing a leaf falling to the ground. Our key innovation is an end-to-end differentiable inverse graphics framework, which jointly models object geometry, physical properties, and interactions directly from videos. Through backpropagation, our approach enables the recovery of force representations from object motions. We validate our method on both synthetic and real-world scenarios, and the results demonstrate its ability to infer plausible force fields from videos. Furthermore, we show the potential applications of our approach, including physics-based video generation and editing. We hope our approach sheds light on understanding and modeling the physical process behind pixels, bridging the gap

---

[*]Equal contribution.

39th Conference on Neural Information Processing Systems (NeurIPS 2025).

between vision and physics. Please check more video results in our project page

*"Who has seen the wind? Neither I nor you: But when the leaves hang trembling, the wind is passing through."*          *– Christina Rossetti*

## 1 Introduction

Watching leaves swirl and glide through the autumn breeze, we can almost sense the wind gently guiding them in a natural choreography. Similarly, as cherry blossom petals drift in spring, it feels as though the air cradles them, orchestrating their delicate descent. Although we cannot directly see the wind, humans can seamlessly infer these *invisible* physical interactions from visible cues in their surroundings, such as those captured in videos. While this intuitive physics capability has long existed in human vision, it remains underexplored in computer vision. In this paper, we bridge the gap by introducing a differentiable framework to revealing invisible forces from visual data.

The key challenge of this problem lies in extracting insights about an unseen target—dynamic forces—while relying exclusively on visible inputs. To address this, it is essential to understand how videos, as visible cues, connect to the underlying invisible dynamics. Consider a video of a leaf falling: external forces like wind, apply to a leaf with known shape, appearance, and physical properties, producing a motion that aligns with physical laws and is captured visually. By developing an end-to-end differentiable model of this physical process, we can learn and predict these invisible forces, such as wind, based on video evidence alone.

To this end, we propose a differentiable inverse graphics framework, which models objects' inherent properties (geometry, appearance, and physical properties), invisible force representations, and physical processes from video inputs. For object modeling, we leverage 3D Gaussians [1] as representations for shape and appearance, which can be easily obtained from videos. To model objects' physical properties, we propose a novel approach that leverages commonsense about physical properties in vision-language models and attaches the knowledge to 3D Gaussians. For force representations, we adopt the Eulerian perspective and introduce a novel causal tri-plane representation, which models the spatio-temporal continuity and intrinsic causality of forces with high fidelity. For physical processes, we implement a differentiable physics simulator for deformable objects to animate object motions based on object properties and forces. We note that our object representation (Gaussians as Lagrangian elements) and the force representation (causal tri-plane as grids) perfectly fit into the formulations of the material point method [2], allowing us to accurately model the physical process. Together, these components form a differentiable framework that bridges perception and physics, so that we can estimate forces from video object motions via backpropagation.

While our framework accurately models the physical process, recovering force representations from object motions in videos remains highly challenging. Unlike system identification approaches which estimate only a few physical parameters, forces are omni-directional and can present throughout the 3D space. Estimating such dense and complex force representations poses great challenges to optimization. Moreover, time integration in the physics simulator leads to unstable backpropagation, with gradients often exploding as they accumulate over time. To address the challenges, we propose a novel 4D sparse tracking objective, where we represent object motions as the movements of sparse keypoints in the spatio-temporal space, and the movements of the Lagrangian elements, i.e., the 3D Gaussians, are further controlled by their neighboring keypoints via barycentric interpolation. With this objective, we greatly reduce the complexity of the prediction space and facilitate the estimating of force representations.

We evaluated our estimated force representations on both synthetic and real-world scenarios. The results demonstrate our method's ability to recover invisible forces from videos. Moreover, we show that with the estimated force representation, we can generate novel and physically plausible object motions by changing object types, physical properties, or boundary conditions, which enables realistic physics-based video generation and editing.

To conclude, we summarize our contributions as follows:

- We identify an important problem in physics understanding from videos: recovering invisible forces from object motions. To tackle this problem, we propose a novel inverse-graphics framework

that jointly models object properties, forces, and physical processes, enabling the estimation of underlying forces directly from video observations.

- We introduce a novel sparse tracking objective, which effectively handles the optimization challenges in differentiable physics and enables robust estimation of forces from visual inputs.

- We demonstrate our method's ability to recover forces from motion, and showcase its potential for generating physically plausible motions and enabling physics-based video generation and editing.

## 2    Related Works

**Intuitive Physics.** Understanding the physical world is a fundamental aspect of human intelligence. Researchers have long sought to bring this intuitive physics understanding ability to machine intelligence. Galileo [3] and the following works [4, 5] integrated deep learning with physics simulation to estimate physical object properties from visual observations. More recent approaches performed system identification by leveraging differentiable physics [6–9], neural fields [10–12], 3D Gaussian splatting [13, 14], vision-language models [15, 16], and video generation [17–20], enabling more accurate estimation of physical object properties. However, these methods primarily focus on a single physical parameter, such as mass, friction, and Young's modulus. In contrast, estimating forces is significantly more challenging, as they are vectors that can exist throughout the 3D space. In this paper, we propose a novel framework that successfully recovers force fields from visual inputs.

**Differentiable Physics.** Differentiable physics simulators [21–29] have been widely used to bridge perception and physics by enabling the backpropagation of particle motion gradients to physical parameters. However, using gradients from physics simulators to optimize physical properties can be notoriously difficult, as the inherent discontinuous behavior and the time integration of physics simulation often lead to vanishing or exploding gradients. To handle this challenge, we propose an optimization scheme with a novel sparse tracking objective, which greatly stabilizes the estimation process and enables robust recovery of high-dimensional forces.

**Force Estimation.** Researchers explored modeling contact forces for robotic manipulation [30–34] and human-object interactions [35–40]. However, most approaches rely on controlled robotic environments with tactile sensors or require strong priors on hand and object shapes, as well as physical properties, to estimate forces. In contrast, our method operates on natural videos with minimal assumptions about object properties, enabling force estimation in unconstrained scenarios.

**Physics-based Generation.** Researchers explored reconstructing physically interactive scenes [41, 42] and generating physically plausible videos [43–46]. Most approaches rely on physics simulators or physics-informed neural networks [47] to animate motion, but they typically require manually specified forces and environmental conditions. Beyond these, interactive editing methods [48, 49] drive visual changes by optimizing displacement fields in the image or feature space under generative priors; such formulations specify apparent motion without estimating underlying physical forces. An alternative approach learns 3D velocity fields directly from videos [50, 51], producing smooth trajectories yet lacking explicit force representations, which makes parameter-aware edits (e.g., changing mass) less principled. In contrast, our approach automatically recovers forces and physical conditions from natural videos and applies them to novel objects, enabling physics-driven video generation without manual parameter tuning.

## 3    Method

We study recovering invisible forces from videos. Our inverse graphics framework first models object properties (Section 3.1), force representations (Section 3.2), and physical processes (Section 3.3) from videos. To optimize force representations, we introduce a sparse tracking objective (Section 3.4). An overview of our method is in Figure 2.

### 3.1    Object Modeling

Capturing the essence of dynamic objects requires modeling both their shape for accurate physical interactions and their appearance for visual fidelity. This necessitates a representation that seamlessly integrates precise Lagrangian shape modeling with photorealistic rendering. To this end, we adopt 3D Gaussians [1] as the representation for shape and appearance. Specifically, an object in a video is

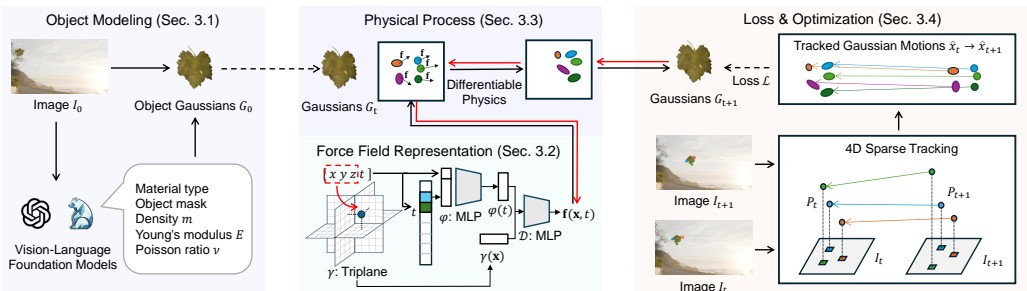

Figure 2: We propose a differentiable inverse graphics framework to recover invisible forces from videos by integrating object modeling, physics simulation, and optimization. Objects are represented with 3D Gaussians and assigned physical properties via Vision-Language Models. Forces are modeled as a causal tri-plane, and object motions are simulated using a differentiable physics simulator. A sparse tracking objective enables robust differentiable force recovery from videos.

represented by a set of Gaussian kernels $G$ in the 3D space. Each Gaussian kernel $G$ is parameterized by

$$G = \{\mathbf{x}, \mathbf{v}, \boldsymbol{\Sigma}, \sigma, SH, \mathbf{D}, m, E, \nu\}, \tag{1}$$

where $\mathbf{x} \in \mathbb{R}^3$ and $\mathbf{v} \in \mathbb{R}^3$ are the spatial location and velocity of a Gaussian kernel respectively. The covariance matrix $\boldsymbol{\Sigma}$ represents the shape, and opacity $\sigma$ and spherical harmonics $SH$ represent the appearance of a Gaussian. Moreover, we attach each Gaussian with its physical properties: a deformation gradient $\mathbf{D}$, the mass $m$, Young's modulus $E$, and the Poisson ratio $\nu$, which we will discuss later.

Since objects in a video undergo motion and deformation due to external forces, their corresponding 3D Gaussians also evolve over time. Let $t$ denote a timestep in the video. The Gaussians at time $t$ is then defined as

$$G^t = \{\mathbf{x}^t, \mathbf{v}^t, \boldsymbol{\Sigma}^t, \sigma, SH, \mathbf{D}^t, m, E, \nu\}, \tag{2}$$

where the spatial position $\mathbf{x}^t$, velocity $\mathbf{v}^t$, covariance $\boldsymbol{\Sigma}^t$, and deformation gradient $\mathbf{D}^t$ change over time $t$.

We initialize the Gaussians $\{G^0\}$ at $t = 0$ only using the first frame of the video. Specifically, we use pixel-aligned point clouds that are extracted from the first image $I^0$ via a pretrained metric-depth model [52] to initialize the Gaussian positions $\{\mathbf{x}^0\}$, and we optimize $\{\boldsymbol{\Sigma}^0, \sigma, SH\}$ via Gaussian splatting on $I^0$. Notably, although point clouds from a single image can be incomplete, we can still obtain robust force estimates thanks to the proposed sparse tracking objective, which will be discussed later. We have also explored multiview object reconstruction in our experiments. For $\mathbf{v}^0$ and $\mathbf{D}^0$, we initialized them as $\mathbf{0}$ and $\mathbf{I}$ respectively.

To model the physical interactions between objects and forces, we also need to know the objects' physical properties from videos. To this end, we introduce a simple but effective approach that leverages commonsense knowledge from vision-language models to assign the physical properties $\{m, E, \nu\}$ to each 3D Gaussian. Specifically, given the first image $I^0$, we first query a vision-language model [53] to infer the object types and provide an estimate of the physical properties $\{m, E, \nu\}$ from commonsense knowledge. Then, we query a grounded segmentation model [54] to generate object segmentation masks based on the object types. Finally, the pixel-aligned Gaussians $\{G^0\}$ that are inside the object masks are assigned with the corresponding estimated physical properties $\{m, E, \nu\}$. For common objects, the estimated physical properties from the vision-language model are quite robust. Hence, even without accurate system identification, our framework could provide a robust force estimate with the commonsense physical properties(see Experiments 4.3).

Leveraging foundation models [52–54], our framework automatically recovers objects' geometry, appearance, and physical properties from videos without manual effort. The recovered object Gaussians serve as a unified representation for modeling physical interactions in dynamic videos.

## 3.2 Force Representations

Properly modeling force representations is essential to our framework. For point contact forces, we can directly define force vectors on Gaussian particles. However, for forces that are distributed throughout 3D space, *e.g.*, wind, we adopt the Eulerian perspective and introduce a causal tri-plane to represent forces in 3D space. This representation is based on the observations that forces are spatially continuous and causally dependent over time. Specifically, we define the force $\mathbf{f}$ at the position $\mathbf{x}$ and the time $t$ as

$$\mathbf{f}(\mathbf{x}, t) = \mathcal{D}(\gamma(\mathbf{x}) + \varphi(t; \varphi(t-1))), \tag{3}$$

where $\mathcal{D}(\cdot)$ is a feature decoder and $\gamma(\cdot)$ represents the tri-plane feature map from [55]. $\varphi(\cdot)$ is a small MLP that encodes the time $t$, initialized using the learned weights from the previous timestep $t-1$, *i.e.*, $\varphi(t-1)$. Compared to other 4D representations [56–58], this representation disentangles space and time, leading to superior computational efficiency. Additionally, the recursive dependency of $\varphi(t)$ on $\varphi(t-1)$ enables accurate modeling of evolving force dynamics over time.

## 3.3 Physical Process

With the object Gaussians and force representations, we are ready to simulate object motion following physical laws. To this end, we implemented a differentiable physics simulator for deformable objects using the Material Point Method (MPM) [2]. Empirically, we found that our object representation, where Gaussians act as Lagrangian elements, and our force representation, modeled as a causal tri-plane on a grid, naturally align with the formulations of [2], enabling accurate modeling of the physical process.

In detail, the forward physical process $F_{physics}$ takes the object Gaussians $\{G^t\}$ at time $t$ and the force field $\mathbf{f}(\mathbf{x}, t)$, and outputs $\{G^{t+1}\}$ at the next timestep $t+1$:

$$\{G^{t+1}\} = F_{physics}(\{G^t\}, \mathbf{f}(\mathbf{x}, t)). \tag{4}$$

The physical process relies on multiple sub-steps $\delta t$ to update motions from $t$ to $t+1$ incrementally. In the following section, we introduce the computational flow in a sub-step $\delta t$. For simplicity, we consider a single Gaussian $G^t$ and omit the particle-to-grid, grid computation, and grid-to-particle process in MPM, focusing solely on the core physics principles and update formulas.

For a Gaussian $G^t : \{\mathbf{x}^t, \mathbf{v}^t, \mathbf{\Sigma}^t, \sigma, SH, \mathbf{D}^t, m, E, \nu\}$ at the time $t$, we first characterize the object deformations by updating the deformation gradient $\mathbf{D}^t$:

$$\mathbf{D}^t = (\mathbf{I} + \nabla\mathbf{v}^{t-\delta t}\delta t)\mathbf{D}^{t-\delta t}, \tag{5}$$

where $\nabla\mathbf{v}^{t-\delta t}$ is the velocity gradient and $\mathbf{I}$ is the identity matrix. Then, we update the Gaussian velocity $\mathbf{v}^t$ by incorporating both external and internal forces:

$$\mathbf{v}^t = \mathbf{v}^{t-\delta t} + \delta t \frac{\mathbf{f}(\mathbf{x}^t, t)}{\mathbf{m}} + \delta t \frac{\mathbf{f}_i(\mathbf{x}^t, t, E, \nu, \mathbf{D}^t)}{\mathbf{m}}, \tag{6}$$

where $\mathbf{f}(\mathbf{x}^t, t)$ is the external force by querying the casual tri-plane $\mathbf{f}(\mathbf{x}, t)$ at the Gaussian position $\mathbf{x}^t$, $\mathbf{f}_i(\mathbf{x}^t, t, E, \nu, \mathbf{D}^t)$ represents internal forces $\mathbf{f}_i$ determined by the constitutive model. $\mathbf{m}$ is the mass matrix derived from the mass $m$ of the Gaussian. Note that external forces are computed on particles before the particle-to-grid step, while internal forces are calculated on the corresponding grid during the grid-to-particle step. External forces are applied directly to particles because the scene volume is much larger than the occupied regions. Acting on particles instead of grid nodes avoids empty cells and yields finer, less noisy fields aligned with the moving mass.

Next, we update the position $\mathbf{x}^t$ of the Gaussian following standard time integration:

$$\mathbf{x}^t = \mathbf{x}^{t-\delta t} + \delta t\mathbf{v}^t. \tag{7}$$

Finally, the covariance matrix $\mathbf{\Sigma}$ is calculated based on the deformation gradient:

$$\mathbf{\Sigma}^t = \mathbf{D}^t\mathbf{\Sigma}^0(\mathbf{D}^t)^T. \tag{8}$$

Through multiple sub-step updates, we can evolve the Gaussian state from $G^t$ to $G^{t+1}$. Notably, the computational process is fully differentiable. Hence, given the per-Gaussian motion $\hat{\mathbf{x}}^t \rightarrow \hat{\mathbf{x}}^{t+1}$

extracted from adjacent video frames, we leverage $\hat{\mathbf{x}}^t \rightarrow \hat{\mathbf{x}}^{t+1}$ as the motion tracking target to optimize the force field $\mathbf{f}(\mathbf{x}, t)$ via backpropagation.

$$\min_{\mathbf{f}(\mathbf{x},t)} |\hat{\mathbf{x}}^{t+1} - \mathbf{x}^{t+1}|,$$
$$\text{s.t.} \quad (\hat{\mathbf{x}}^t, \mathbf{x}^{t+1}) \in (G^t, G^{t+1}), \tag{9}$$
$$G^{t+1} = F_{physics}(G^t, \mathbf{f}(\hat{\mathbf{x}}^t, t)).$$

In the following sections, we will discuss how to establish per-Gaussian motion $\hat{\mathbf{x}}^t \rightarrow \hat{\mathbf{x}}^{t+1}$ from videos, and how to optimize the force field $\mathbf{f}(\mathbf{x}, t)$ robustly.

### 3.4 Recovering Forces from Videos

To optimize the force field $\mathbf{f}(\mathbf{x}, t)$, it is essential to track per-Gaussian motions $\hat{\mathbf{x}}^t \rightarrow \hat{\mathbf{x}}^{t+1}$ from videos as the optimization target. A straightforward approach is to use a photometric loss, *i.e.* comparing the pixel differences in adjacent frames $|I^t - I^{t+1}|$ by projecting the Gaussians onto the image plane: $I^t = \pi(\{G^t\})$, where $\pi$ is the projection function. Nevertheless, we found that photometric loss alone fails to provide sufficient motion constraints, often resulting in vanishing gradients during optimization. Alternatively, we can extract dense 3D scene flows from videos using off-the-shelf depth and optical flow prediction or 4D reconstruction [59–61]. Nevertheless, the accuracy of these pre-trained models is limited, resulting in noisy dense 3D flows that significantly hinder the optimization process. We found the key to robust optimization is to reduce the target space and adopt more reliable motion estimates. To this end, for objects with bending-only deformations (e.g., paper folding) or small deformations, we introduce a novel 4D sparse-tracking objective. Specifically, we adopt a more reliable point-tracking algorithm [62] that provides sparse, pixel-level estimates of object keypoint motions $\mathbf{p}^t \rightarrow \mathbf{p}^{t+1}$, where $\mathbf{p} \in \mathbb{R}^{N \times 2}$ is $N$ keypoint pixel coordinates. Next, we want to establish the keypoint correspondences in 3D: $\mathbf{P}^t \rightarrow \mathbf{P}^{t+1}$, where we use $\mathbf{P} \in \mathbb{R}^{N \times 3}$ and $\mathbf{p} \in \mathbb{R}^{N \times 2}$ to denote the associated keypoints in the 3D and pixel space respectively. To obtain $\mathbf{P}^t \rightarrow \mathbf{P}^{t+1}$ from $\mathbf{p}^t \rightarrow \mathbf{p}^{t+1}$, we first estimate the 3D keypoint locations $\mathbf{P}^0$ in the first frame by unprojecting $\mathbf{p}^0$ into 3D with depth estimates $d$:

$$\mathbf{P}^0 = \pi^{-1}(\mathbf{p}^0, d), \tag{10}$$

where $d$ comes from a metric-depth model [52] and $\pi^{-1}$ is an inverse-projection function. Then, for each adjacent frames, we obtain $\mathbf{P}^t \rightarrow \mathbf{P}^{t+1}$ from $\mathbf{P}^t$ and $\mathbf{p}^{t+1}$ by optimizing the following objective:

$$\min_{\mathbf{P}^t \rightarrow \mathbf{P}^{t+1}} |\pi(\mathbf{P}^{t+1}) - \mathbf{p}^{t+1}| + \lambda \mathcal{L}_{arap}, \tag{11}$$

where the as-rigid-as-possible loss $\mathcal{L}_{arap}$ is represented as

$$\mathcal{L}_{arap} = \Sigma_{i,j \in \mathbf{P}} |(\mathbf{P}_i^{t+1} - \mathbf{P}_j^{t+1}) - (\mathbf{P}_i^t - \mathbf{P}_j^t)|. \tag{12}$$

By minimizing the re-projection errors while keeping the object skeleton as rigid as possible, we obtain a robust estimate of the 3D keypoint motions $\mathbf{P}^t \rightarrow \mathbf{P}^{t+1}$. Notably, without reliance on per-frame depth estimation, our method circumvents the inconsistent video depth estimation problem and enables more robust 3D motion estimates.

Next, we leverage the sparse keypoint motions $\mathbf{P}^t \rightarrow \mathbf{P}^{t+1}$ to control the per-Gaussian motions $\hat{\mathbf{x}}^t \rightarrow \hat{\mathbf{x}}^{t+1}$ via

$$\hat{\mathbf{x}} = \alpha_i \mathbf{P}_i + \alpha_j \mathbf{P}_j + \alpha_k \mathbf{P}_k, \tag{13}$$

where the Equation 13 is the barycentric interpolation, $P_{i,j,k}$ are the 3-nearest neighbors of $\hat{\mathbf{x}}$, and the coefficients $\alpha_{i,j,k}$ are computed in the first frame and fixed in the following frames. The sparse keypoints $\mathbf{P}$ characterize the object skeletons and control the fine-grained Gaussian positions $\hat{\mathbf{x}}$. Compared to the 3D scene flow approach that directly tracks each Gaussian's motion, estimating sparse keypoint motions $\mathbf{P}^t \rightarrow \mathbf{P}^{t+1}$ reduces the prediction space and demonstrates superior robustness and accuracy, allowing us to obtain high-quality $\hat{\mathbf{x}}^t \rightarrow \hat{\mathbf{x}}^{t+1}$ for optimizing the force field.

Finally, we optimize the force $\mathbf{f}(\mathbf{x}, t)$ using the motion-tracking loss $\mathcal{L}_{motion}$ in Equation 9, with the estimated Gaussian motions $\hat{\mathbf{x}}^t \rightarrow \hat{\mathbf{x}}^{t+1}$. Moreover, we add two regularization terms $\mathcal{L}_{space}$ and $\mathcal{L}_{time}$ for spatial and temporal smoothness, respectively:

$$\mathcal{L} = \mathcal{L}_{motion} + \lambda_1 \mathcal{L}_{space} + \lambda_2 \mathcal{L}_{time}, \tag{14}$$

where $\mathcal{L}_{space}$ follows [56] and penalize the total variation in space, and $\mathcal{L}_{time}$ encourages temporal smoothness by penalizing the parameter differences of the time encoder $\varphi(\cdot)$:

$$\mathcal{L}_{time} = |\varphi_\theta^{t+1} - \varphi_\theta^t|, \tag{15}$$

where $\varphi_\theta^t$ is the parameters of the time encoder $\varphi$ at the time $t$ in Equation 3. With the losses in Equation 14, we are able to recover the force $\mathbf{f}(\mathbf{x}, t)$ by tracking the Gaussian motions $\hat{\mathbf{x}}^t \rightarrow \hat{\mathbf{x}}^{t+1}$ extracted from videos.

## 4    Experiments

In this section, we conduct comprehensive experiments to investigate the following key questions:

- Can our method successfully recover forces from both synthetic and real-world videos? (Section 4.2)

- How do the proposed components, *i.e.*, force representation and loss function affect the final performance, and how robust is the proposed VLM framework to variations in object physical properties?(Section 4.3)

- How can our method be applied to physics-based video generation and editing? (Section 4.4)

### 4.1    Experimental Setup

We conduct experiments on both real-world and synthetic data. For real-world data, we leverage Internet videos to verify the physical plausibility of recovered forces by visualizing the force field. In addition, we conduct real physical experiments with a force gauge to measure the actual forces, and we evaluate our recovered forces via re-simulation. For synthetic data, we use synthetic objects in [63, 18] to evaluate the numerical accuracy of recovered forces. We leverage objects from 3 distinct material types, *i.e.*, elastic, elastoplastic, viscoplastic, 6-8 unique force fields, and 2 different camera viewpoints to build the synthetic scenarios in the physics simulator [24]. We use rendered videos as inputs to our system to estimate the forces and compare them with the ground truth forces in the simulator. For numerical comparisons in synthetic scenarios, we adopt image reconstruction metrics, *i.e.*, PSNR, SSIM [64], and LPIPS [65], to compare the re-simulated videos with the recovered forces and the original input videos, to demonstrate the accuracy of recovered forces to match the ground truth object motions in simulation. Moreover, we compare the recovered forces with the ground truth forces using two metrics: average magnitude error (reported as percentages) and direction error (measured in degrees).

### 4.2    Force Recovery

**In-the-Wild Videos.** We evaluate our method on real Internet videos to demonstrate its ability to recover plausible force fields from natural object motions. Figure 4 presents qualitative results on various scenes. Our method successfully infers the underlying forces by observing object deformations and trajectories over time. The visualized forces dynamically adapt to object motion, demonstrating a physically plausible force field that varies over time.

**Controlled Real-World Physics Experiments.** Since the ground truth forces in real Internet videos are unknown, we conduct controlled real-world experiments to further validate our method. Using a force gauge, we apply known forces to an object while capturing its motion on video. We then use our method to recover the force field and reapply it to the object in simulation. As shown in Figure 3, our method successfully reconstructs the object's motion and deformation, closely aligning with real-world observations. The experimental results demonstrate the accuracy of our recovered forces in real-world scenarios.

**Synthetic Scenarios.** To evaluate the numerical accuracy of recovered forces, we build synthetic scenarios in the physics simulator to obtain the "ground truth" forces. As shown in Table 1, our method successfully recovers the original force fields with low numerical errors. These quantitative results provide strong evidence that our approach accurately estimates the underlying force dynamics and generalizes well across various objects and physical properties.

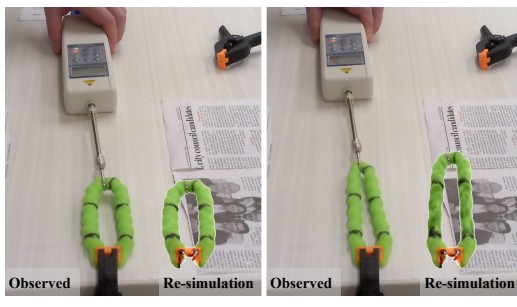

Figure 3: Comparison of observed data (left in each frame) and re-simulated results (right in each frame) for two different frames in the real-world experiment.

| Material Type | Method | PSNR | SSIM | LPIPS | Mag. Error (%) | Dir. Error (°) |
|---|---|---|---|---|---|---|
| Elastic | Point | 20.57 | 0.94 | 0.04 | 95.91 | 76.48 |
| | K-Planes | 26.25 | 0.96 | 0.03 | 18.03 | 39.83 |
| | Ours | **39.79** | **0.99** | **0.01** | **5.14** | **4.38** |
| Elastoplastic | Point | 31.40 | 0.98 | 0.01 | 98.36 | **26.8** |
| | K-planes | 30.14 | 0.98 | 0.01 | 87.81 | 61.42 |
| | Ours | **39.93** | **0.99** | 0.01 | **75.06** | 45.50 |
| Viscoplastic | Point | 17.49 | 0.95 | 0.12 | 98.30 | 92.02 |
| | K-planes | **41.73** | 0.99 | 0.03 | 89.04 | 33.09 |
| | Ours | 39.00 | **0.99** | **0.01** | **21.44** | **7.50** |

Table 2: Quantitative comparison of force representations.

| Loss functions | PSNR | SSIM | LPIPS | Mag. Error(%) | Dir. Error (°) |
|---|---|---|---|---|---|
| Image Loss | 37.24 | 0.99 | 0.01 | 86.74 | 50.23 |
| Flow+Depth Loss | **41.54** | 0.99 | 0.01 | 27.90 | 16.07 |
| Ours | 39.79 | **0.99** | **0.01** | **5.14** | **4.38** |

Table 3: Quantitative comparison of loss functions.

| Material type | Object | PSNR | SSIM | LPIPS | Mag. Error (%) | Dir. Error (°) |
|---|---|---|---|---|---|---|
| Elastic | Lego | 33.70 | 0.98 | 0.01 | 19.53 | 7.02 |
| | Ficus | 25.92 | 0.94 | 0.03 | 23.97 | 11.55 |
| | Sunflower | 34.08 | 0.99 | 0.01 | 14.38 | 7.85 |
| Elastoplastic | Toy | 41.35 | 0.99 | 0.00 | 29.19 | 8.11 |
| | Chair | 40.10 | 0.99 | 0.00 | 33.31 | 23.40 |
| Viscoplastic | Hotdog | 30.63 | 0.96 | 0.02 | 15.09 | 11.63 |

Table 1: Force recovery in synthetic scenarios.

| Material Type | #Samples | Type F1 | $\rho$ MAPE (%) | $E$ log-MAPE(%) |
|---|---|---|---|---|
| Elastic | 6 | 1 | 2.38 | 5.31 |
| Elastoplastic | 6 | 1 | 10.80 | 3.36 |
| Viscoplastic | 5 | 1 | 0 | 13.98 |
| Overall | 17 | 1 | 4.65 | 7.17 |

Table 4: VLM performance on the daily-item dataset.

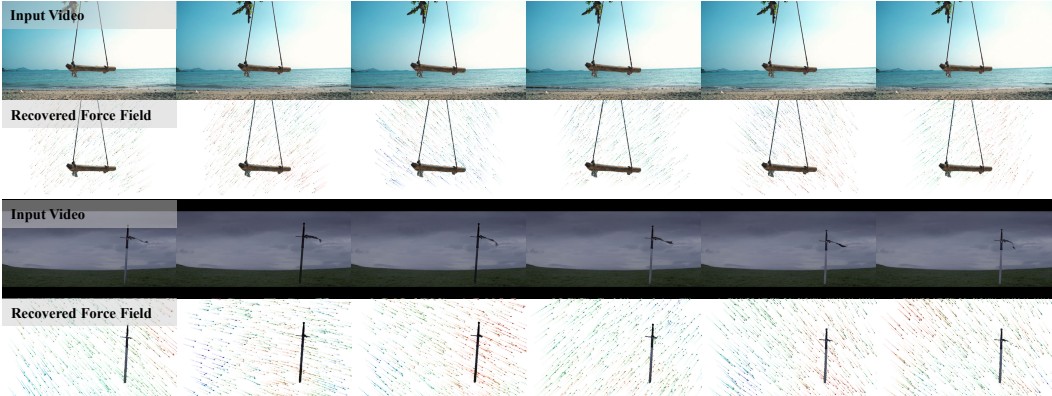

Figure 4: Our method estimates invisible force fields from real-world videos, producing physically plausible motion interpretations.

## 4.3 Empirical Study

**Force Representations.** To evaluate the effectiveness of our force representation in Section 3.2, we compare our causal tri-plane with other 4D representations such as K-planes [56] and point contact forces. The results in Table 2 demonstrate that our causal tri-plane force representation performs better than other 4D representations. This is mainly because our force representation accurately models the spatial continuity and temporal dependence of forces.

**Loss Functions.** To evaluate the effectiveness of our loss functions in Section 3.4, we compare our sparse tracking loss with the image reconstruction loss and the dense 3D scene flow loss derived from depth and flow estimation (Flow+Depth loss). The results in Table 3 demonstrate that our sparse tracking loss shows better performance than others, especially in force accuracy. This is because sparse tracking provides more robust motion estimates that can be leveraged as more accurate signals to optimize the forces.

**VLM Material-Property Estimation.** To evaluate the robustness of leveraging a vision-language model for physical parameter estimation, we measure the estimation errors by utilizing the GPT-4o-Vision to infer material type and material properties(density $\rho$ and Young's modulus $E$) from a single image and comparing with the ground truth values. A small benchmark of 17 everyday objects

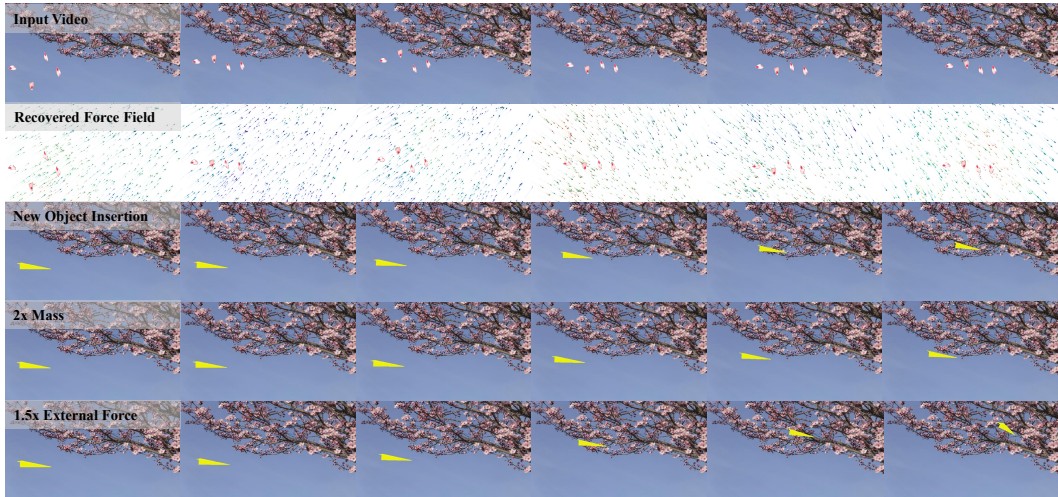

Figure 5: Our method recovers force fields from input videos and enables the insertion of novel objects while maintaining physically plausible motion. We demonstrate new object insertion, and modifications of physical conditions (e.g. mass and external force), showcasing the model's ability to generate physically plausible videos.

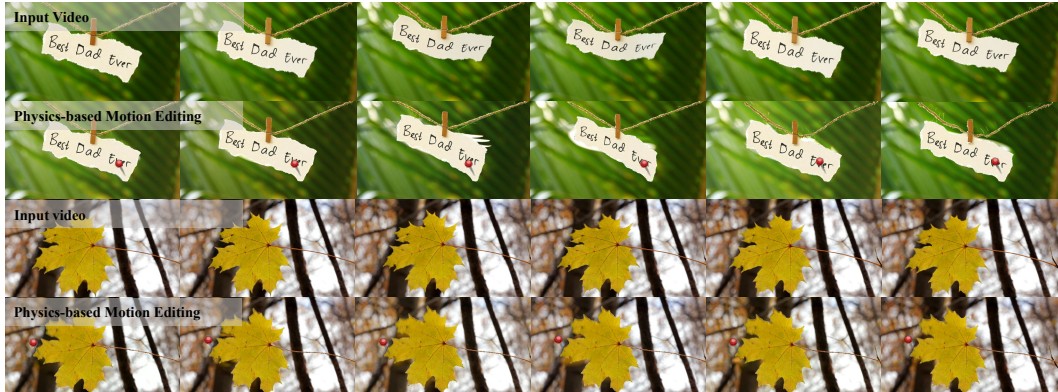

Figure 6: Our method allows modifying object motion by adjusting external constraints while preserving physical realism. We demonstrate how altering boundary conditions (e.g., fixing parts of an object) influences motion under the same estimated force field. These results highlight the flexibility of our approach for controllable, physics-based video editing.

is collected, each made of a single, well-documented engineering material. Ground-truth $\rho$ and $E$ values are taken from MatWeb database[66]. Results are summarised in Table 4. The model achieves an overall F1 of 1.0 for material classification, an $4.65\%$ mean-absolute-percentage error (MAPE) on density, and a log-MAPE of 7.17%, which indicates that the VLM can have a robust initial estimate of the physical parameters.

**Moving Cameras** For non-stationary cameras we adopt 4D reconstruction pipelines to supply camera poses and trajectories, which we consume without retraining. Qualitative videos for dynamic-camera sequences are best viewed on the project page.

### 4.4 Physics-based Generation

With the recovered force field, we demonstrate the potential of our approach for physics-based video generation and editing. Figure 5 shows the results of physics-based video generation. Our framework enables the replacement of novel objects within the same force field, generating physically plausible motions for novel objects via physics simulation. This flexibility also allows us to modify the object's

physical properties and force strengths to create distinct object motions that adhere to physical laws. Compared to other video generation methods, our approach produces more controllable and physically-accurate videos. Figure 6 shows the results of physics-based video editing, where we can modify the boundary conditions in the scenes, *e.g.*, fixing a point, to generate different physically plausible object motions in the same video. These results highlight the versatility of our framework in generating and editing videos while maintaining physical consistency.

We qualitatively compare against interactive editing / motion-driven methods [48, 49] and velocity-field learning baselines [50, 51] on matched inputs. These approaches optimize displacements or velocities under strong priors and thus yield kinematically plausible results, but they do not estimate identifiable physical forces and do not enforce Newtonian consistency when physical parameters are edited (e.g., doubling mass). Consequently, the resulting motion may continue to "match" an appearance prior yet diverge from the dynamics implied by the edited parameters. By explicitly recovering time-varying forces inside a differentiable simulator, our method preserves dynamical consistency under parameter edits and non-uniform fields. Qualitative videos are best viewed on the project page.

## 5 Conclusion

In this work, we introduced a differentiable inverse graphics framework to recover invisible forces from video object motions, bridging vision and physics. By modeling object properties, forces, and physical processes, our method enables robust force estimation via backpropagation. Experiments in both real-world and synthetic data have demonstrated accurate force recovery and controllable physics-based video generation and editing of our approach.

**Limitations.** Our framework is primarily applied to objects with small deformation or bending-only deformation. Fluids or other object types that require different, differentiable physical processes are out of the scope of this paper, which is left for future works.

As in our demo, we model foreground physics and composite over a static background, as off-the-shelf per-frame inpainting can introduce temporal flicker that obscures our contribution.

## Acknowledgments and Disclosure of Funding

This work is in part supported by NSF RI #2211258 and #2338203, ONR MURI N00014-22-1-2740, and the Okawa Research Grant. The USC Geometry, Vision, and Learning Lab acknowledges generous supports from Toyota Research Institute, Dolby, Google DeepMind, Capital One, Nvidia, and Qualcomm. Yue Wang is also supported by a Powell Research Award.

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
