# OpenReview forum: "Seeing the Wind from a Falling Leaf"
_NeurIPS.cc/2025/Conference — NeurIPS 2025 poster_

### Official Review · Reviewer_MYah · 2025-06-23

**Clarity:** 3
**Significance:** 4
**Originality:** 4
**Rating:** 5
**Confidence:** 3

**Summary:**

This paper presents an innovative inverse graphics framework for recovering invisible forces (such as wind) from video data. The key idea is to estimate dense, time-varying force fields by modeling object geometry and physical properties using 3D Gaussians, and simulating dynamics via a differentiable physics engine. A novel causal tri-plane representation encodes spatio-temporal force fields, and a sparse 4D tracking objective enables effective gradient-based optimization. Experiments on both synthetic and real-world data demonstrate strong performance in recovering force fields and simulating physically plausible object behavior.

**Questions:**

- How sensitive is the method to inaccuracies in depth estimation or object segmentation from vision-language models (VLMs) and SAM?
- What directions might improve the realism of the generated and edited videos? Could this method eventually support practical applications such as content creation or simulation?
- Can the framework be adapted for real-time use in mobile or robotics settings, where interactive or reactive force estimation is required?

**Ethical Concerns:**

["NO or VERY MINOR ethics concerns only"]

**Final Justification:**

The paper presents a well-rounded and complete study. The proposed method is novel, and the experimental evaluation is thorough and convincing. The author response addressed my initial concerns adequately. I have decided to keep my original rating (5: Accept).

**Limitations:**

yes

**Quality:**

4

**Strengths And Weaknesses:**

## Strength：

1. The idea of estimating invisible forces purely from object motion is novel, underexplored, and highly compelling.
2. The integration of 3D Gaussians, causal tri-plane force modeling, and differentiable physics is both elegant and technically effective.
3. The proposed optimization strategy is well-motivated and successfully addresses common issues with gradient instability in differentiable simulation. The experimental evaluation is thorough and comprehensive, effectively validating the method’s performance.

## Weaknesses:

- **Limited applicability to object types**: As acknowledged in the paper, the proposed framework currently supports only solid objects with small or bending-only deformations. More complex materials such as fluids or granular substances remain out of scope.
- **Heavy reliance on pretrained models**: The accuracy of the force estimation pipeline depends heavily on initial object geometry and physical property estimations derived from pretrained vision-language and depth models. This reliance could become a bottleneck in more complex or noisy real-world scenarios.
- **Limited realism in generation and editing**: Although the model demonstrates strong performance in force recovery and simulation, the visual quality of the generated and edited videos appears noticeably less realistic compared to recent state-of-the-art video generation models. This limits the perceived quality and potential practical deployment of the method in high-fidelity applications.

---

> ### Author Rebuttal · Authors · 2025-07-31
>
> We are grateful for your thorough and insightful review. Your feedback has highlighted several important aspects, and we will carefully address each of your concerns in our response.
>
> > **Q1**:  How sensitive is the method to inaccuracies in depth estimation or object segmentation from vision-language models (VLMs) and SAM?
>
> **A1**: Our method is generally robust to inaccuracies in depth estimation and object segmentation due to the following reasons.
> As noted in Section 3.1, while object segmentation is important for assigning physical properties to the 3D Gaussians, SAM is generally reliable for this task, and the object-background separation it provides is usually sufficient for our method.
>
> Inaccurate depth estimation can impact tracking, but our sparse tracking objective (described in Section 3.4) effectively handles this issue. By focusing on sparse keypoints rather than relying on dense depth data, we reduce the sensitivity to depth errors, making the method more robust to noise in depth estimation. You can also refer to Table 3 for related results.
>
> Our experiments on in-the-wild video in Section 4.2 show that the method performs well even with minor errors in depth or segmentation. While significant errors can still affect the results, the sparse tracking approach ensures that the force recovery process remains reliable.
>
> In summary, our sparse tracking mechanism effectively mitigates the impact of depth estimation inaccuracies, as demonstrated in our experiments.
>
>
> > **Q2**: What directions might improve the realism of the generated and edited videos? Could this method eventually support practical applications such as content creation or simulation?
>
> **A2**: As demonstrated in Figures 4, 5, and 6, our method has shown potential in content creation and simulation for real-world applications. These examples highlight how realistic object interactions and force fields can be modeled in various scenarios.
>
> To improve realism, future work could refine the physical simulation to handle complex materials, such as fluids or hyper-elastic deformations, and integrate more advanced models for collisions, friction, and lighting effects, enhancing both visual fidelity and physical accuracy.
>
> This method could support practical applications like content creation and simulation. For example, in film production, it could simulate realistic environmental interactions, such as wind, while in virtual reality, it could enable physics-driven interactions. It could also be used in simulation for robotics and industrial design, where accurate object behavior under forces is critical.
>
> > **Q3**: Can the framework be adapted for real-time use in mobile or robotics settings, where interactive or reactive force estimation is required?
>
> **A3**: Our framework can be applied to offline robotic data to estimate physical interactions, and with this information we can train a robotic policy that can be deployed in the real world. For instance, accurate wind-force estimation is valuable for pneumatic non-prehensile “blowing’’ manipulation, where a mobile blower herds scattered objects into a receptacle[1]. We will investigate this direction in the future. Thank you for the advice!
>
> > **Q4**: Limited applicability to object types: As acknowledged in the paper, the proposed framework currently supports only solid objects with small or bending-only deformations. More complex materials such as fluids or granular substances remain out of scope.
>
> **A4**: Our current implementation is indeed limited to solid objects exhibiting only small or bending-dominant deformations; fluids and granular media remain beyond its scope. However, it can be addressed within our framework, albeit with a particle-based differentiable physical process. We will include more discussion in the limitation section of our paper and plan to address it in future iterations of the framework, where we hope to expand its applicability to a wider range of materials.
>
> > **Q5**: Heavy reliance on pretrained models: The accuracy of the force estimation pipeline depends heavily on initial object geometry and physical property estimations derived from pretrained vision-language and depth models. This reliance could become a bottleneck in more complex or noisy real-world scenarios.
>
> **A5**: As discussed in Q1 and Section 3.4, our framework is designed to be resilient to errors in these initial estimates. The sparse tracking objective focuses on sparse, reliable keypoints for motion tracking. This approach helps mitigate the potential bottleneck caused by errors in the pre-trained models, particularly in real-world scenarios where noise and complexity may arise.
>
> That being said, we agree that further improvements in the robustness of pre-trained models, especially in noisy or complex environments, would benefit our approach. We continue to explore methods to refine the initial object geometry and physical property estimations in future iterations of the framework.
>
>
> > **Q6**:  Limited realism in generation and editing: Although the model demonstrates strong performance in force recovery and simulation, the visual quality of the generated and edited videos appears noticeably less realistic compared to recent state-of-the-art video generation models. This limits the perceived quality and potential practical deployment of the method in high-fidelity applications.
>
> **A6**: Currently, the visual quality is constrained by the limitations of neural rendering techniques, but this can be addressed by integrating diffusion models to rectify the artifacts from neural rendering or using them as motion guidance to control a video diffusion model, similar to approaches like Difix3D[2]. This will be explored in future work.
>
> [1] Wu, Jimmy, et al. "Learning pneumatic non-prehensile manipulation with a mobile blower." IEEE Robotics and Automation Letters 7.3 (2022): 8471-8478.
>
> [2] Wu, Jay Zhangjie, et al. "Difix3d+: Improving 3d reconstructions with single-step diffusion models." Proceedings of the Computer Vision and Pattern Recognition Conference. 2025.

---

> > ### Comment · Reviewer_MYah · 2025-08-01
> >
> > Thank you for the detailed and thoughtful response. I appreciate the clarifications provided, which addressed my concerns. I will keep my original rating.

---

> > > ### Author Response · Authors · 2025-08-02
> > >
> > > Thanks for your feedback! If all the concerns have been addressed, we would really appreciate it if you could consider **raising your score**. If you have more questions, feel free to let us know during the rebuttal window!

---

### Official Review · Reviewer_PKhM · 2025-06-24

**Clarity:** 3
**Significance:** 3
**Originality:** 3
**Rating:** 4
**Confidence:** 4

**Summary:**

This paper presents an end-to-end framework that combines differentiable simulation with differentiable rendering to extract an implicit force field from video observations. To stabilize the optimization of the force field, a sparse tracking objective is utilized. Experiments conducted on both synthetic and real-world videos demonstrate the effectiveness of the force estimation process. Additionally, the learned implicit force field can be generalized to new configurations, such as inserting new objects or increasing the magnitude of the force.

**Questions:**

1.	Several existing studies [a-b] have demonstrated the ability to estimate a velocity field from video observations, which is analogous to the proposed method that estimates a force field (assuming constant mass). It is recommended that the authors discuss the differences between these studies and the proposed method. Additionally, they should present quantitative comparisons to determine which approach yields more effective results.

2.	It is challenging to reconstruct the 3D scene given only the first frame of the entire video. How does the method handle scenes that include object occlusions? Could the rest frames provide some information for reconstructing the 3D scene?

3.	In Section 3.1 of the manuscript, the authors mainly discuss the modeling of foreground objects. However, it is unclear whether the background is modeled in the same way as the foreground objects. If that is the case, how does the method obtain the physical parameters (e.g., mass and Young's modulus) for the background particles? Additionally, in the supplementary videos, while the original clips show noticeable background motion, the videos generated through physics-based motion editing do not display any background movement. Can the authors provide some clarification on this?

4.	In Section 3.3 of the manuscript, what is the consideration for applying external forces on each particle before the particle-to-grid step, rather than applying them on each grid node, similar to internal forces? For Equation 13, have the authors considered alternative interpolation strategies (e.g., K nearest neighbors), and how do they perform? What is the ratio between the number of $\mathbf{x}$ and $\mathbf{P}$?

5.	The examples presented in the paper primarily involve force fields that have similar directions and a fixed input view. Can the proposed method address the following scenarios: (a) cases where the force directions are distinct (for instance, two leaves float from the left and right edges of the image toward the center, influenced by opposing forces) and (b) input videos that capture both camera movements and object movements?

References

[a] NVFi: Neural Velocity Fields for 3D Physics Learning from Dynamic Videos. NeurIPS 2023.

[b] FreeGave: 3D Physics Learning from Dynamic Videos by Gaussian Velocity. CVPR 2025.

**Ethical Concerns:**

["NO or VERY MINOR ethics concerns only"]

**Final Justification:**

Most of my initial concerns have been addressed by the authors' response. While the experiments primarily demonstrate simple motions, which might be seen as a limitation of the study, I believe they could inspire future research in this field. Therefore, I tend to uphold my original rating.

**Limitations:**

Yes

**Paper Formatting Concerns:**

No paper formatting concerns

**Quality:**

3

**Strengths And Weaknesses:**

This paper introduces a unified framework for estimating implicit force fields using only video observations. It presents a novel force representation that disentangles space and time with recursive dependency. Experiments demonstrate the framework's effectiveness in extracting force fields and applying the learned fields to new configurations. Despite its strengths, it also has some noticeable weaknesses, e.g., considering only very simple motions as inputs and failing to make comparisons with existing works. Overall, this work explores an intriguing direction by combining differentiable physics with video understanding, and it may inspire future research in similar areas.

---

> ### Author Rebuttal · Authors · 2025-07-31
>
> We sincerely appreciate your valuable comments and suggestions. Your feedback has helped us refine our work, and we are excited to clarify and discuss each of your points in the following.
>
> > **Q1**: Several existing studies [a-b] have demonstrated the ability to estimate a velocity field from video observations, which is analogous to the proposed method that estimates a force field (assuming constant mass). It is recommended that the authors discuss the differences between these studies and the proposed method. Additionally, they should present quantitative comparisons to determine which approach yields more effective results.
> > References
> > [a] NVFi: Neural Velocity Fields for 3D Physics Learning from Dynamic Videos. NeurIPS 2023.
> > [b] FreeGave: 3D Physics Learning from Dynamic Videos by Gaussian Velocity. CVPR 2025.
>
> **A1**: The key difference between our method and the mentioned NVFi and FreeGave is that **these methods don’t rely on explicit and differential physical processes** to model physical interactions. Specifically, these methods rely on a neural network that mimics the physical interactions and predicts particle velocities, with a physics-based regularization term to constrain the outputs. In contrast, our method leverages an explicit physical process to model the physical interactions and learn the force field via a differential physics engine. **In the revision, we will add a dedicated discussion of both NVFi and FreeGave to clarify these distinctions.** Thank you!
>
>
> > **Q2**:It is challenging to reconstruct the 3D scene given only the first frame of the entire video. How does the method handle scenes that include object occlusions? Could the rest frames provide some information for reconstructing the 3D scene?
>
> **A2**: Our method relies on the first frame to initialize the 3D reconstruction of the scene, as described in Section 3.1. The 3D Gaussians representing the objects are initialized using pixel-aligned point clouds extracted from the first image via a pre-trained metric-depth model. This initialization is performed once using the first frame and does not get updated by subsequent frames.
>
> While the first frame provides the essential geometric initialization, we also mention in Section 3.1 that we have explored **both single-view (line 122) and multi-view (line 127) reconstruction techniques**. In the case of multi-view input, additional perspectives can further refine the 3D model, but the geometry itself is still primarily initialized from the first frame.
>
> Regarding object occlusions, the method handles them through sparse tracking of keypoints, which allows for robust motion tracking even in the presence of partial occlusion. By focusing on key points that are less likely to be occluded, we can maintain the accuracy of the 3D reconstruction and force estimation despite object occlusions.
>
>
> > **Q3**: In Section 3.1 of the manuscript, the authors mainly discuss the modeling of foreground objects. However, it is unclear whether the background is modeled in the same way as the foreground objects. If that is the case, how does the method obtain the physical parameters (e.g., mass and Young's modulus) for the background particles? Additionally, in the supplementary videos, while the original clips show noticeable background motion, the videos generated through physics-based motion editing do not display any background movement. Can the authors provide some clarification on this?
>
> **A3**: In our framework, we specifically model **only the foreground objects** and not the background. The background is treated as a static element in our simulation. The physical properties such as mass and Young’s modulus are assigned only to the foreground objects, and background is not modeled with these physical parameters.
>
> Regarding the lack of background motion: for all demos we assume the background is static. We extract the background from the first frame, inpaint occluded regions, and then composite our simulated foreground motion on top. This choice keeps the focus on foreground dynamics and sidesteps the additional complexity of modeling background physics.
>
> We have indeed experimented with segmenting and inpainting every frame, then compositing our simulated objects into those reconstructed backgrounds. Current off-the-shelf inpainting techniques, however, still exhibit minor temporal flicker across frames, which can distract from our main contribution. For clarity of presentation, we therefore adopt the simpler static-background pipeline in the demos, allowing the reader to focus on the foreground physics.
>
>
> > **Q4**: In Section 3.3 of the manuscript, what is the consideration for applying external forces on each particle before the particle-to-grid step, rather than applying them on each grid node, similar to internal forces? For Equation 13, have the authors considered alternative interpolation strategies (e.g., K nearest neighbors), and how do they perform? What is the ratio between the number of x and P?
>
> **A4**: We chose to apply external forces to each particle rather than to the grid nodes to achieve finer force sampling. Because the scene volume is large compared with the objects, only a small fraction of grid cells are occupied in any given frame; applying forces at grid nodes would therefore leave most cells with zero force, yielding a coarse and noisy field.
>
> By applying forces to particles, we can achieve more accurate and detailed force estimations, as the forces are directly linked to the moving particles, allowing for better resolution and precision.
>
> Regarding Equation 13, we selected barycentric interpolation because it works well for structured data like skeleton-driven animation. It ensures smooth transitions between keyframes, which is essential for our setup involving small deformations or bending-only deformations. While KNN interpolation could be useful for more irregular or unstructured data, it may result in less smooth transitions and be computationally expensive when dealing with large datasets.
>
> In our experiments, the number of particles x typically ranges around 10k to100k, while the number of P (keypoints) is on the order of 10, so the ratio of x to P is generally in the range of 1k to 10k.
>
>
> > **Q5**: The examples presented in the paper primarily involve force fields that have similar directions and a fixed input view. Can the proposed method address the following scenarios: (a) cases where the force directions are distinct (for instance, two leaves float from the left and right edges of the image toward the center, influenced by opposing forces) and (b) input videos that capture both camera movements and object movements?
>
> **A5**: For Scenario A (distinct force directions): Our method is **already capable of handling situations where forces act in different directions**. In our real-world experiments, such as those involving in-the-wild videos, we demonstrated that the framework can successfully model forces that vary across the scene, e.g., in Figures 1 and 4. The force field is estimated at each point in the scene, allowing for distinct directions of force acting on different objects simultaneously. This flexibility in force direction handling has been validated in our experiments with random forces applied to objects, as shown in our results.
>
> For Scenario B (camera and object movements): Our model **doesn’t assume a static camera**. As long as we can obtain a robust estimation of object motions in 4D, our method can infer forces from the motions. With the recent advances in 4D reconstruction and camera pose estimation, such as Shape of Motion[1] and MegaSAM[2], our method can be naturally extended to moving cameras.
>
> [1] Wang, Qianqian, et al. "Shape of motion: 4d reconstruction from a single video." arXiv preprint arXiv:2407.13764 (2024).
>
> [2] Li, Zhengqi, et al. "MegaSaM: Accurate, fast and robust structure and motion from casual dynamic videos." Proceedings of the Computer Vision and Pattern Recognition Conference. 2025.

---

> > ### Comment · Reviewer_PKhM · 2025-08-02
> >
> > Thank you for your detailed response. While my major concerns have been addressed, I still recommend including an experiment that compares NVFi/FreeGave with the proposed approach. Since these methods are all aimed at estimating force/velocity fields from observed videos, this additional experiment could enhance the current manuscript. It would demonstrate the effectiveness of explicit physics in comparison to learning-based physics-constraint methods.

---

> > > ### Author Response · Authors · 2025-08-02
> > >
> > > Thanks for your response! We agree with your point that it would be good to compare with learning-based methods. Due to the rebuttal regulations, we are not able to provide a qualitative comparison at this moment, but we will add this comparison in our future revisions.
> > >
> > > We once again thank your efforts in reviewing our paper! Your valuable comments really help improve our paper, and inspire us a lot! We hope our response has addressed your concerns. It'd be great if you could also consider **raising your score** as well. If you have more questions, don't hesitate to let us know in the rebuttal window!

---

> > > > ### Comment · Reviewer_PKhM · 2025-08-04
> > > >
> > > > Thanks for the response. Please make sure to include the rebuttal content in the revised paper.

---

> > > > > ### Author Response · Authors · 2025-08-04
> > > > >
> > > > > Thank you for the response! Yes, we will make sure to include all the rebuttal content in the final paper.

---

### Official Review · Reviewer_Utuv · 2025-07-09

**Clarity:** 4
**Significance:** 3
**Originality:** 4
**Rating:** 4
**Confidence:** 3

**Summary:**

This paper aims to infer unseen forces (mostly wind) from real video. To do so, they construct an inverse graphics pipeline that takes into account the geometry of objects in the scene as well as their motion through time. This inverse graphics pipeline is differentiable, so can be optimized with gradient descent. This pipeline first extracts details about the objects in the scene using multimodal LLMs and segmentation to get info about material types and object boundaries. Then, modeling outcomes of inferred force fields as gaussians, they can optimize for the force field which produced the observed video. They demonstrate applications such as object insertion and force field modification.

**Questions:**

Please look at weaknesses. In general, I'm inclined to vote for acceptance due to the creative setting/solutions. The reason my score is not higher is because the scale of the results/quality of the generations seems like they limit broader applicability. I also think a comparison to recent interactive simulation work is very important (e.g. DragGAN and "Motion Prompting: Controlling Video Generation with Motion Trajectories")

**Ethical Concerns:**

["NO or VERY MINOR ethics concerns only"]

**Final Justification:**

I am inclined towards acceptance, however the lack of more complex scenes/examples preclude a higher score. I understand that theoretically, this framework can be extended to more complex scenes. However the devil is in the details when dealing with video inference/generation, so I would need to see the execution. Unfortunately, the format of the rebuttal does not allow for this. Hopefully I can see these examples in the final revision.

**Limitations:**

See above

**Paper Formatting Concerns:**

No concerns

**Quality:**

3

**Strengths And Weaknesses:**

Strengths
+ Very creative idea. Inferring forces from video has been generally studied but I haven't not seen this specific approach that allows for deformable materials.
+ Intuitive strategy. The paper is written in a way that the rather complex inverse rendering pipeline is fairly intuitive.
+ Insertion results are interesting. They demonstrate that, given an inferred force field, they can insert new objects that obey those forces. This seems generally applicable in other fields for things like data augmentation.

 Weaknesses
- Missing discussion on the substantially related interactive simulation line of work (e.g. DragGAN). This is very important. The authors should show when those methods fail relative to this one, as they can also infer motion through optical flow and allow for insertion/editing.
- The "in-the-wild" qualitative results provided are entirely focused on wind, which limits the scope of applicability. There could be other unseen forces, e.g. gravity/object occluded actions. Also potentially interested in the elasticity results for in-the-wild data.
- Qualitatively, the new object insertion results are strange. In particular, for the paper airplane, the leaves demonstrate a slight upward lift close to the start of the video which is not reflected in the airplane's trajectory. As such, I would appreciate some more uncurated examples for the rebuttal.
- All results/demo videos are done with respect to short video and a static camera. With the existing pipeline, it seems difficult to extend many of the results to real video with dynamic cameras (which could be important for broader applicability).

---

> ### Author Rebuttal · Authors · 2025-07-31
>
> Thank you for your comprehensive and constructive feedback. We truly appreciate the time you spent reviewing our paper and are grateful for your suggestions, which we will address point by point.
>
> > **Q1**: Missing discussion on the substantially related interactive simulation line of work (e.g. DragGAN). This is very important. The authors should show when those methods fail relative to this one, as they can also infer motion through optical flow and allow for insertion/editing.
>
> **A1**: DragGAN and Motion Prompting are relevant to our work. The key difference is that **these methods do not follow an explicit physical law**. These methods infer motion through optical flow, using the term "force" to represent a direction of motion, which is more akin to displacement than to actual physical forces. In contrast, our approach explicitly models the underlying physical interactions—such as wind, gravity, and interaction forces—that govern the motion and deformation of objects. This distinction is crucial because our method estimates real physical forces, enabling more accurate and physically plausible simulations, especially in complex scenarios where precise force fields need to be recovered. **In the revision, we will add a dedicated discussion of both papers and include a qualitative comparison to highlight these distinctions.**
>
>
>
> > **Q2**: The "in-the-wild" qualitative results provided are entirely focused on wind, which limits the scope of applicability. There could be other unseen forces, e.g. gravity/object occluded actions. Also potentially interested in the elasticity results for in-the-wild data.
>
> **A2**: First, it’s important to note that our framework does **indeed account for gravity** in the scenarios, as gravity is inherently included in the physical processes modeled in our system. This is particularly evident in how we capture and simulate object motion under natural conditions, including the effects of gravity on objects in the scene.
>
> Regarding **elasticity and handling object occluded actions**, the results shown in Figure 1, particularly with the sticker example, demonstrate that our method is capable of handling objects with significant deformation and occlusion. The sticker example illustrates the model’s ability to track and simulate forces on objects undergoing elastic deformations, even in the presence of partial occlusion.
>
> In summary, while the qualitative results presented focus on wind, our approach is capable of handling a broader range of forces, as demonstrated by the sticker example and other scenarios.
>
>
> > **Q3**: Qualitatively, the new object insertion results are strange. In particular, for the paper airplane, the leaves demonstrate a slight upward lift close to the start of the video which is not reflected in the airplane's trajectory. As such, I would appreciate some more uncurated examples for the rebuttal.
>
> **A3**: Thank you for noting that the leaves rise slightly at the start, whereas the inserted paper airplane stays nearly level. This difference in behavior is due to the **different physical properties** of the two objects, such as mass and aerodynamic characteristics, which affect how they respond to the forces applied in the simulation. The leaves, being lighter and more affected by the wind, show a noticeable upward lift, while the heavier paper airplane reacts differently to the same forces.
>
> > **Q4**: All results/demo videos are done with respect to short video and a static camera. With the existing pipeline, it seems difficult to extend many of the results to real video with dynamic cameras (which could be important for broader applicability).
>
> **A4**: Our model **doesn’t assume short video or a static camera**. As long as we can obtain a robust estimation of object motions in 4D, our method can infer forces from the motions. With the recent advances in 4D reconstruction and camera pose estimation, such as Shape of Motion[1] and MegaSAM[2], our method can be naturally extended to moving cameras.
>
> [1] Wang, Qianqian, et al. "Shape of motion: 4d reconstruction from a single video." arXiv preprint arXiv:2407.13764 (2024).
>
> [2] Li, Zhengqi, et al. "MegaSaM: Accurate, fast and robust structure and motion from casual dynamic videos." Proceedings of the Computer Vision and Pattern Recognition Conference. 2025.

---

> > ### Comment · Reviewer_Utuv · 2025-08-06
> > **Response**
> >
> > I thank the author for their considered rebuttal. I am still inclined towards acceptance. The author rebuttal has partially addressed my comments. In particular Q4, while I understand that the framework in theory could be applied to more complex data with non-stationary camera, in practice this has not been shown in existing examples/evals. I will decide on a final score during the reviewer discussion period.

---

> > > ### Author Response · Authors · 2025-08-08
> > >
> > > Thank you for highlighting the importance of demonstrating results under moving-camera conditions. Due to the short rebuttal period and the NeurIPS restriction on adding new visualizations, our current submission includes only static-camera examples. However, the framework itself is inherently camera-motion agnostic, provided that accurate 4D trajectories are available. In the final version, we will incorporate qualitative results on dynamic-camera sequences and add a focused discussion on integrating our method with recent 4D reconstruction pipelines such as Shape-of-Motion and MegaSAM. We believe these additions will directly address your concerns and further strengthen the paper.
> > >
> > > We sincerely appreciate the time and insight you have invested in reviewing our work. Your feedback has already helped us refine the manuscript and has sparked several new experiments. We hope this response resolves your remaining questions and, if so, would be grateful if you would **consider a higher score**. Please feel free to reach out with any further comments during the discussion period!

---

### Official Review · Reviewer_mwjJ · 2025-07-21

**Clarity:** 4
**Significance:** 1
**Originality:** 4
**Rating:** 5
**Confidence:** 4

**Summary:**

In this paper the authors present a new model for inferring a force field of “wind” by analyzing videos. This is performed by tying together a differentiable physics engine with sparse point tracking (along side a VLM for property estimation). This method recovers known forces in natural and synthetic videos, produces qualitatively reasonable vector fields for natural videos of objects flapping in the wind, and allows for video editing by changing known properties (e.g., making an object twice as heavy or the wind twice as strong)

**Questions:**

1. As mentioned above, the only real-world application presented for this model is the ability to edit or generate videos. Are there other applications for this work that might be of use to the broader NeurIPS or computer vision community?
2. All of the examples in this paper present videos with a single focus object (e.g. one leaf or a swing). Would this model work with multiple objects? With particulate matter (e.g. snow)? Or is the object modeling limited to a single thing?
3. A VLM is used in the object identification phase to identify material properties to add latent physical properties, and against a small benchmark provides good property estimation. But how important is this to the larger model – e.g., if you assume only average information, or a misidentification (e.g. something that looks like cloth but acts like cardboard) how brittle is the full model?
4. What types of forces are used in the synthetic video set? Are the uniform in magnitude and direction but time varying? Or can the wind be blowing in two directions at once? Looking at Figure 4, the force field appears to be pointing in the same direction for each frame, and the forces appear to have a smooth gradient across the frame (generally peaking around the object) so it’s not clear whether this is just because the examples focus on a single object with wind where local direction and magnitude estimates are most important, or whether this is a heuristic found by the model

**Ethical Concerns:**

["NO or VERY MINOR ethics concerns only"]

**Final Justification:**

The authors appropriately addressed my (and other reviewers') comments. I believe it would be nice to have an explicit test of the effect of the VLM on model performance, but this is a minor point and does not affect my overall rating. Thus my rating (Accept) stands

**Limitations:**

yes

**Quality:**

3

**Strengths And Weaknesses:**

This paper is clear and well written, and presents an interesting and novel model for inferring physical forces from videos, especially working on soft bodies. However, its main weakness is that it is not clear what this algorithm could be used for beyond the video editing examples presented in this paper – which make up a small part of video editing and generation as a whole – and thus it is not clear how broad the audience for this work would be beyond those finding inspiration in the algorithm for other purposes

---

> ### Author Rebuttal · Authors · 2025-07-31
>
> We would like to express our sincere gratitude for your thoughtful and detailed feedback. Your comments have provided valuable insights, and we are eager to address them in detail.
>
> > **Q1**: As mentioned above, the only real-world application presented for this model is the ability to edit or generate videos. Are there other applications for this work that might be of use to the broader NeurIPS or computer vision community?
>
> **A1**: Beyond video generation and editing, one of the potential applications of our work is in **augmented and virtual reality (AR/VR)**, where our method can model physics-driven behaviour of virtual objects. This could enable more immersive AR/VR experiences by dynamically adjusting objects' movements based on simulated forces like gravity or wind, which could be particularly useful in interactive gaming environments.
>
> Another potential application is in **robotics and simulation**. By estimating the forces involved in object interactions, our framework could enhance robotic manipulation in dynamic environments—for example, guiding pneumatic non-prehensile “blowing’’ systems that corral objects into a receptacle [1]. This could improve robots’ ability to handle deformable or flexible items by modelling the forces acting on them, leading to better motion planning and control.
>
> [1] Wu, Jimmy, et al. "Learning pneumatic non-prehensile manipulation with a mobile blower." IEEE Robotics and Automation Letters 7.3 (2022): 8471-8478.
>
>
> > **Q2**: All of the examples in this paper present videos with a single focus object (e.g. one leaf or a swing). Would this model work with multiple objects? With particulate matter (e.g. snow)? Or is the object modeling limited to a single thing?
>
> **A2**: Our model **does support multiple objects**. For example, Figure 1 illustrates a scenario with four stickers, each modeled individually, and Figure 5 shows four sakura petals being modeled independently. The pipeline allows for separate modeling of each object’s geometry, physical properties, and interactions, making it compatible with multi-object scenarios by treating each object as an independent entity while still modeling the overall force field acting on them.
>
> Regarding particulate matter like snow, as we mentioned in our limitation section, we acknowledge that our model is primarily designed for objects with small or bending-only deformation. That said, our method can model particulate matter with a particle-based differentiable physical process; this is beyond the scope of this work and we leave it for future exploration.
>
>
> > **Q3**: A VLM is used in the object identification phase to identify material properties to add latent physical properties, and against a small benchmark provides good property estimation. But how important is this to the larger model – e.g., if you assume only average information, or a misidentification (e.g. something that looks like cloth but acts like cardboard) how brittle is the full model?
>
> **A3**: We found that if the physical parameters are in a reasonable range, our framework is quite robust to object identification. Since VLM contains commonsense knowledge about object physical properties, we found it is sufficient for our use. A misidentification in the case you referred to may lead to a failure in our model, but in our experiments, we didn’t notice such a severe hallucination of VLMs.
>
>
> > **Q4**: What types of forces are used in the synthetic video set? Are the uniform in magnitude and direction but time varying? Or can the wind be blowing in two directions at once? Looking at Figure 4, the force field appears to be pointing in the same direction for each frame, and the forces appear to have a smooth gradient across the frame (generally peaking around the object) so it’s not clear whether this is just because the examples focus on a single object with wind where local direction and magnitude estimates are most important, or whether this is a heuristic found by the model
>
> **A4**: Thank you for your inquiry! Firstly, for a clear and fair quantitative evaluation of our method, we set the forces to be time-varying but spatially uniform in our synthetic data.
>
> Secondly, yes, the wind can indeed be blowing in two directions at once. This is because we are capturing the force field at each time step, and the force field is a result of the combined forces acting on the object. In scenarios where multiple forces are acting simultaneously, they are captured as a cumulative force field, which can have different directions and magnitudes at different spatial locations in the scene.
>
> Regarding the observations in Figure 4, indeed the force field appears to be pointing in the same direction for each frame with a smooth gradient. This apparent uniformity is specific to that example and not a systematic tendency of our model’s estimates. In other examples, such as those in Figure 1, the force field is more complex and varies in both direction and magnitude, highlighting that the method is capable of handling more diverse scenarios, not just uniform forces.

---

### Decision · Program_Chairs · 2025-09-17

**Decision:**

Accept (poster)

**Comment:**

This work is aimed at recovering the physical force field from input videos. This is achieved using a differentiable inverse graphics framework which can model multiple objects which can undergo time-varying small bending deformations. Metric depth prediction models, object segmentation, and multi-modal LLMs are used to initialize the object depth, boundary, and physical properties like mass and Young's modulus. A tracking objective based on sparse key point tracking (using co-tracker) is also used, which helps with occlusions. Applications show video generation with object insertion, and force field modification.

The reviewers appreciated the novelty and creativity of the approach, esp. in modelling deformable objects, and were had positive ratings for this work. There were a few concerns regarding comparisons to prior (learning based) works, and, limited application domain, e.g., wrt moving cameras. In view of the above, I'm inclined to recommend this work for acceptance. I encourage the authors to address the limitations highlighted by the reviewers and incorporate their valuable feedback.